# High-Temperature Properties of LP-DED Additive Manufactured Ferritic STS 430 Deposits on Martensitic STS 410 Base Metal

**DOI:** 10.3390/mi16050494

**Published:** 2025-04-23

**Authors:** Samsub Byun, Hyun-Ki Kang, Namhyun Kang, Seunghun Lee

**Affiliations:** 1R&D Center, Turbo Power Tech, #107 Dasan-ro, Saha-gu, Busan 49488, Republic of Korea; ssbyun333@gmail.com; 2Department of Materials Science and Engineering, Pusan National University, Busandaehak-ro 63beon-gil, Geumjeong-gu, Busan 46241, Republic of Korea; 3Welding & Joining Development Team, TESONE Co., Ltd., #37 Nakdong-daero 550beon-gil, Saha-gu, Busan 49315, Republic of Korea; sh_lee@tes-one.com

**Keywords:** additive manufacturing, directed energy deposition, ferritic stainless steel 430, martensitic stainless steel 410, chrome carbide precipitation

## Abstract

The aim of this work is to study the phase transformations, microstructures, and mechanical properties of ferritic stainless steel (FSS) 430 deposits on martensitic stainless steel (MSS) 410 base metal (BM) using laser powder-directed energy deposition (LP-DED) additive manufacturing. The LP-DED additive manufactured FSS 430 deposits on MSS 410 BM underwent post-heat treatment at 815 °C and 980 °C for 1 h, respectively. The analyses of phase transformations and microstructural evolutions of LP-DED FSS 430 on MSS 410 BM were carried out using X-ray diffraction, SEM, and EBSD. The highest strain was observed at the coarsened chromium carbide (Cr_23_C_6_) in the joint interface between AM FSS 430 and MSS 410 MB. This contributed to localized lattice distortion and mismatch in crystal structure between chromium carbide and the surrounding ferrite. Tensile strength properties at elevated temperatures were discussed to investigate the effects of the different post-heat treatments. The tensile properties of the as-built samples including tensile strength of about 550 MPa and elongation of about 20%, were the same as those of the commercial FSS 430 material. Tensile properties at 500 °C indicated a modest increase in tensile strength to 540–550 MPa. The specimens heat treated at 980 °C retained higher tensile strength than those heat treated at 815 °C. This would be attributed to the grain refinement from prior LP-DED microstructure and chromium carbide coarsening at higher heat treatment, which can increase dislocation density and yield harder mechanical behavior.

## 1. Introduction

Directed energy deposition (DED) additive manufacturing is a type of digital manufacturing processes from a CAD model which produces a near-net shape or final shape parts composed of complicated geometries using a focused energy source such as a laser, electron beam, or plasma arc to melt and materials deposited onto a surface layer by layer [1,2,3,4]. For metal additive manufacturing (AM) processes, laser-based heat sources have been widely used for powder bed fusion (PBF) and directed energy deposition (DED) processes due to their capability to achieve precise, localized, and high-energy input for focusing metal powders, enabling the formation of complex geometries with high precision controls [5,6,7].

Compared to austenite stainless steels such as AISI 304 and AISI 316, ferritic stainless steel 430 has disadvantages, including environmentally lower corrosion resistance, poor weldability, susceptibility to embrittlement, limited work hardening ability, and unsuitable mechanical properties for high-strength applications. Therefore, the uses of ferric stainless steel (FSS) 430 are limited to kitchen appliances [8,9], automotive (trim, moldings, wheel covers, and exhaust systems) [10,11,12], and architectural applications [13,14,15]. Nevertheless, FSS 430 has received some great attention because it possesses good corrosion resistance in mild environments and is a significantly more cost-effective choice compared to Ni-bearing austenitic stainless steels [16,17,18]. In particular, FSS 430 has been commonly utilized in steam turbine seals such as labyrinth seals [19,20], with the advantages of their good corrosion resistance, machinability, and mechanically suitable performance for elevated operating temperatures. The labyrinth seal of FSS 430 plays an important role as a type of non-contact mechanical seal that significantly reduces fluid leakage between rotating and the stationary parts by producing a tortuous path through a complex maze-like pattern of grooves, but it requires good corrosion resistance and a proper mechanical property without damaging the rotor [21,22,23].

Additive manufacturing joint FSS 430 onto MSS 410 may present unique metallurgical and mechanical challenges due to differences in thermal expansion coefficients, phase transformation behaviors, and mechanical properties [24,25,26]. Especially, MSS 410 has characteristics of high strength and hardness, but it may be prone to brittleness and cracking when exposed to rapid heating and cooling cycles [27,28,29,30,31,32]. It has been revealed that the rapid solidification and thermal cycling during the LP-DED processes have a significant impact on the microstructure evolutions, phase transformations, residual stresses, and interfacial bonding between the deposit and the base metal [33,34]. Razzaq et al. [35] reported a comprehensive review on joining dissimilar metals by additive manufacturing, highlighting dissimilar welding challenges of large heat-affected zones (HAZ) and intermetallic compounds at the interfaces, leading to untimely failure and/or cracking. Khodabakhshi et al. [36] studied dissimilar metals deposition by DED based on powder feed laser additive manufacturing with AISI 410L and 316L, suppressing the formation of brittle intermetallic compounds and decreasing the level of thermal stress leading to cracking. Several more research works on additive manufacturing ferritic stainless steels, including AISI 430 onto austenitic stainless steels, have been carried out [37,38,39].

However, there have been no studies on the joint properties of LP-DED additive manufactured ferritic stainless steel 430 deposits on martensitic stainless steel 410 base metals. Therefore, this study attempts to investigate the microstructural evolutions, grain growth and carbide precipitation in the vicinity of the interface, and high-temperature tensile properties of LP-DED additive manufactured FSS 430 deposits on MSS 410 base metals with various post-heat treatments.

## 2. Materials and Methods

### 2.1. Materials and Additive Manufacturing

The integrated LP-DED equipment (AM Solutions Co., Daejeon, Republic of Korea) of a CNC machining center assembled with IPG Photonics (Oxford, MA, USA) ytterbium fiber laser of 1000 W and beam size of 1.2 mm was used for manufacturing additively the FSS 430. The feedstock material of the FSS 430 powder was manufactured using vacuum induction gas atomization (VIGA) process by Shanghai Truer Technology Co., Ltd. (Shanghai, China). For specimen fabrication, a powder with particle sizes of 45–150 μm, flowability of 19.65 g/cm^3^, apparent density of 4.06 g/cm^3^, and tap density of 4.55 g/cm^3^ was employed, as shown in Figure 1.

The chemical compositions of the powder and its deposit are indicated in Table 1, confirming compliance with the standard specification of AISI 430. The composition of the deposit was measured by SPECTROLAB S (San Diego, CA, USA). For the preliminary experiments seen in Figure 2a, we conducted several attempts to find optimized conditions due to the difficulties in constructing the additive manufactured longish cubes before producing mechanical characterization specimens.

Figure 2b depicts the schematic of the AM FSS 430 deposit cube on the MSS 410 base metal for the microstructural observations. Figure 2c illustrates the drawing of the extended FSS 430 deposition along the longitudinal direction on the MSS 410 base metal. Figure 2d illustrates the precise dimensions of the tensile specimen, in which the center of gage length is the interface between the AM FSS 430 deposit and MSS 410 base metal after machining the joined sample, as shown in Figure 2a,c.

Finally, the optimized LP-DED process parameters obtained from the preliminary experiments are listed in Table 2. AISI 410 martensitic stainless steel with 16 mm × 16 mm × H45 mm was used as a base metal as well as a substrate to evaluate the joint properties with the AM FSS 430 using the laser beam deposition.

The laser deposition strategy was developed with bidirectional scanning, alternating laser scanning path by 90° layer by layer to reduce the anisotropy associated with both the microstructures and mechanical properties of the deposition planes, as seen in Figure 3.

### 2.2. Post-Heat Treatment

The samples of the LP-DED FSS 430 (13 mm × 13 mm × H45 mm) deposits on the MSS 410 base metal (16 mm × 16 mm × H45 mm) underwent heat treatment to characterize their microstructures, phase transformations, and mechanical properties and were compared with as-built samples, as seen in Figure 4. Two kinds of heat treatments were applied. The first one was solution heat treated at 815 °C for 1 h at a heating rate of 4.5 °C/min followed by forced fan cooling down to 100 °C at cooling rate of 30 °C/min. The other was solution heat treated at 980 °C for 1 h at a heating rate of 4.5 °C/min followed by forced fan cooling down to 100 °C at cooling rate of 27 °C/min to relieve the residual stress after rapid cooling from the LP-DED FSS 430 onto the MSS 410 base metal.

### 2.3. Characterization of As-Built and Post-Heat-Treated Samples

Two kinds of samples were prepared for the characterization of the LP-DED FSS 430 on MSS 410 base metal, as seen in Figure 2. As-built and post-heat treatment samples were mainly cut into 5 pieces for the tests of X-ray diffraction and microstructure, as seen in Figure 5. X-ray diffraction (XRD) analyses of as-built and post-heat treatment specimens were carried out using an in situ X-ray diffractometer (EMPYREAN, Malvern Panalytical Co., Malvern, UK) with Cu Kα radiation. A continuous mode with step size of 0.013°, a scan rate 0.05°/s, 2θ ranging from 30° to 90°, and power of 30 mA at 40 kV were applied to investigate phase transformation on planes perpendicular (XY) to the longitudinal build direction (Z). The LP-DED samples were also cut into planes parallel (YZ) to the longitudinal build direction (Z) for the microstructure observation at two interfaces between AM FSS 430 and MSS 410 base metal, as seen in Figure 5c.

The metallographic samples for scanning electron microscopy (SEM) were prepared by polishing with a 1 μm diamond suspension and then etched with a Vilella’s reagent (Picric Acid 1 g + ethanol 100 mL + HCl 5 mL). Subsequently, electrolytic polishing was conducted in a solution (Methyl alcohol 90 mL + perchloric acid 10 mL) at 28 volts for 90 s to achieve a refined surface state for electron back scatter diffraction (EBSD) analysis. The microstructures were characterized by SEM and EBSD. Tensile tests for as-built and post-heat treatment specimens were performed at a crosshead speed of 0.125 mm/min and temperatures of 500 °C, respectively, in accordance with the specifications of round bar specimen in the ASTM E8 standard [40] using a universal test machine (Instron 5881).

## 3. Results and Discussion

### 3.1. Phase Analysis of the Deposit

Figure 6 shows the binary Fe-C phase diagram of LP-DED FSS 430 with 17.3 wt.% Cr and 0.0159 wt.% C, which was composed by calculating using Thermo-Calc 2024a with the input chemical composition of the deposit shown in Table 1.

Although Alizadeh-Sh et al. [41] and Lu et al. [42] reported a phase diagram of AISI 430 with a composition of 16.9Cr-0.28Si-0.48Mn-0.05C, revealing the transformation of L → δ-ferrite → γ-austenite → α-ferrite + Cr_23_C_6_, the present phase diagram shows that the LP-DED FSS 430 material began to transform in an instant to ferrite from the liquid state, as the LP-DED FSS 430 deposit rapidly solidified to the temperature of approximately 882.6 °C from above the liquidus temperature at a carbon content of 0.0159 wt.%. It highlights that there is no austenite phase transformation like in the martensite stainless steel AISI 410 and other ferritic stainless steels [26,43,44]. In the overall solidification, ferrite and metal carbide are predominant. Metal carbides began to precipitate below temperatures of about 880 °C. It seems that ferrite, metal carbide (Cr_23_C_6_), and sigma phases appeared and became stable below 640 °C.

The XRD analyses of FSS 430 samples with powder, as-built, and heat-treated deposit at 980 °C for 1 h followed by forced fan cooling are shown in Figure 7. The three major ferrite peaks with BCC structures (110), (200), and (211) were observed in all of the samples. The FSS 430 feedstock powder has the isotropic peak intensities due to the rapid quenching from the typical vacuum induction gas atomization (VIGA) process, where argon gas blows the metal melt during the VIGA process. Compared to the peaks of the feedstock powder, strong α(110) diffraction peaks occurred due to preferred crystallographic orientation by directional solidification, and some α(200) and α(211) peaks appeared due to texture evolution from thermal cycles for long deposition operation during the DED process. The lattice parameters of the BCC ferrite phase were calculated from the XRD data using the (110), (200), and (211) reflections, as shown in Table 3.

The as-built LP-DED FSS 430 sample exhibited an average lattice parameter of 2.874 Å, whereas the sample heat-treated at 980 °C showed a slightly reduced value of 2.873 Å. During heat treatment at 980 °C, Cr-rich carbides such as Cr_23_C_6_ are likely to precipitate along grain boundaries, leading to chromium depletion from the ferrite matrix. This reduction in chromium content results in a slight lattice contraction of the BCC ferrite structure. Notably, a sigma phase (σ) was not detected, as shown in Figure 6, which led to embrittlement, poor corrosion resistance, and difficulty in machining. This is because both the as-built and solution heat treatment at 980 °C for 1 h followed by forced fan cooling avoided the temperature range between 550 °C and 850 °C [44].

### 3.2. Microstructure Behavior of Deposits with Respect to Post-Heat Treatment

Figure 8 shows the microstructural evolutions of LP-DED FSS 430 under various post-heat treatments. For the as-built sample, irregular ferrite grains were observed as the primary microstructure, as shown in Figure 8a, and many tiny chromium carbide precipitates (Cr_23_C_6_) were seen throughout the matrix, as shown in Figure 8d. The larger chromium carbide precipitates were intensively investigated at the grain boundaries, as shown in Figure 8d,g. This implies that chromium with a body-centered cubic has limited solubility in ferrite and segregates to the grain boundaries, precipitating chromium-rich carbides due to its strong affinity for carbon rather than other elements during rapid solidification driven from well-known additive manufacturing processes. Moreover, the sequentially repeated heating and cooling cycles when operating multi-layer deposition develop the chromium carbide coarsening [45,46]. Figure 8b,e show the microstructures of solution heat-treated sample at 815 °C, depicting that the chromium carbide precipitates (Cr_23_C_6_), as seen in Figure 8g, were partially dissolved into the ferrite matrix. Many small Cr_23_C_6_ are observed in the grain along the grain boundaries, as shown in Figure 8h.

Figure 8c depicts that the solution heat-treated specimen at 980 °C, which completely dissolved the prior chromium carbides into the ferrite matrix, led to significant grain growth and coarser chromium carbide re-precipitates or Ostwald ripening [47,48,49], along the grain boundaries, as shown in Figure 8c,f,i. This was due to the high temperature exposure and intermediate cooling rate compared to water cooling. It is noticed that Ostwald ripening is a diffusion-driven coarsening process where larger particles (carbides or precipitates) glow at the expense of smaller ones because smaller ones have a higher surface energy, becoming thermodynamically less stable and over time, atoms diffuse from smaller ones to larger ones, finally leading to coarsening of precipitates or Ostwald ripening.

Figure 9 provides insights into the interface evolutions between the LP-DED FSS 430 and MSS 410 base metal under various heat treatments. Fully ferrite microstructures were observed throughout the matrix, as shown in Figure 9a–f. For the as-built sample, the interface between the AM FSS 430 and heat-affected zone (HAZ) showed irregular dendritic and columnar grain growth extending from the fusion boundary. Most chromium carbides precipitated along the grain boundaries, and coarse dendritic and columnar ferrite grains also occurred, as shown in Figure 9a,d. Fine ferrites were observed at the interface between the HAZ and MSS 410 base metal for the as-built sample, as shown in Figure 9g,j. Meanwhile, most carbides, illustrated by red arrows, were agglomerated and precipitated within the martensite boundaries, as shown in Figure 9p.

However, for the solution heat treatment at 815 °C, newly formed tiny linear dotted line chromium carbide precipitates, illustrated by arrows (Figure 9e), appeared, and the grain boundaries became more distinct after dissolving the prior chromium carbide precipitates. Figure 9k shows that finer chromium carbide precipitates, illustrated by red arrows, appeared due to faster cooling compared to the interface between the AM FSS 430 and HAZ (Figure 9a,d). Tempered martensite and somewhat homogeneous spherical chromium carbide precipitates, illustrated by red arrows, were observed in the substrate (Figure 9q).

For the solution heat treatment at 980 °C, compared to Figure 9a,d, significant chromium carbide coarsening or Ostwald ripening was preferentially developed along the grain boundaries due to long exposure at the high temperature of 980 °C, depleting chromium in the surrounding ferrite grains, as shown Figure 9c,f, which can lead to sensitization, making the material more susceptible to intergranular corrosion [50]. Further developed coarse chromium carbide precipitates or Ostwald ripenings, illustrated by the red letter “A”, were observed, as shown in Figure 9l, which depleted chromium in the ferrite matrix, reducing corrosion resistance. For the MSS 410 base metal at the solution heat treatment at 980 °C, fully developed tempered martensite was observed (Figure 9i).

Figure 10 shows the EBSD phase maps: IQ maps, IPF maps, and KAM maps of parallel (YZ) planes to the building direction of the LP-DED FSS 430 on the MSS 410 base metal with the as-built, SHT-815, and SHT-980 samples. In the as-built interface, finer ferrite grains with an average grain size of 155 μm were observed compared to the SHT-815 and SHT 980 specimens seen in Figure 10d, and generally, the deposited material phase was identified as ferrite alone, as shown in Figure 10g,h,i. However, the grain size increased with solution heat treatments at 815 °C and 980 °C, respectively, due to the grain growth and recrystallization at these temperature exposures. Numerous chromium carbides precipitated unevenly along the grain boundaries for the solution treated at 980 °C for 1 h followed by forced fan cooling because the solution heat treatment at 980 °C sufficiently promoted significant diffusion of carbon and chromium elements. In addition, the KAM value (Figure 10) increased with heat treatment due to more local misorientation in the vicinity of grain boundaries occupied by chromium carbide precipitates compared to the as-built sample (Figure 10j) and post-heat-treated one at 815 °C (Figure 10k). The highest misorientation (KAM = 0.37°) with the solution heat treatment at 980 °C is mostly attributed to significant carbide coarsening. It seems that compared to furnace cooling, the forced fan cooling was relatively faster and led to somewhat coarser carbide precipitation of Cr-rich carbides (Cr_23_C_6_) at the grain boundaries and within grains, as shown in Figure 10c,f,i, while it prevented the deleterious sigma phase formation, which could form if cooled too slowly from the high temperature range of 600~900 °C.

Figure 11 shows the EBSD images of the interfaces for the LP-DED AM FSS 430 on MSS 410 base metal with various heat treatments. In general, diffusion-driven transition zones with partial martensitic transformation, due to repeated thermal cycles followed by a high cooling rate and chromium carbide precipitation, were observed, as shown in Figure 11a–f. Compared to the AM FSS 430 deposit chemical composition in Table 1, MSS 410 contains a higher carbon content ~0.15 wt.% and lower chromium content 11.5~13.4 wt.%. Thus, chromium diffused into MSS 410, and carbon diffused into FSS 430 during the LP-DED process, which built accumulated heat in the deposits. In the as-built interface, finer grains of average grain size of 4.2 μm were observed, as shown in Figure 11d, compared to the SHT-815 and SHT-980 specimens. Meanwhile, the grain size increased to 6.2 μm, and chromium carbide precipitation intensively occurred near the interface for the solution heat-treated at 815 °C for 1 h followed by forced fan cooling. However, compared to SHT-815, the grain size decreased to 5.8 μm, and the chromium carbide precipitates grew into the FSS 430 deposit more deeply for the solution heat-treated at 980 °C for 1 h followed by forced fan cooling, which implies that carbide precipitation causes the depletion of Cr in the surrounding matrix to be susceptible to corrosion, especially for pitting and intergranular corrosion. Generally, most strain occurred in the vicinity of joint interface between the AM FSS 430 deposit and the MSS 410. For the as-built sample, more strain was concentrated beneath the joint interface and heat-affected zone, resulting in a high KAM value of 0.97°. However, in the SHT-815 interface, most strain was concentrated along the interface, but not in the HAZ due to the thermal gradient during forced fan cooling. The highest strain was observed at the coarsened chromium carbides above the interface, beneath the joint interface, and in the heat-affected zone. Regarding the KAM values, it must be noticed that the chromium carbide precipitates (Cr_23_C_6_) are prone to exhibit a higher KAM value than the ferrite matrix because these precipitates create localized lattice distortions. In addition, the mismatch in crystal structure between the chromium carbide and the surrounding ferrite matrix can introduce high internal stresses. It seems that chromium carbides act as barriers to dislocation movement and cause dislocations to pile up at their interfaces. As a result, the accumulation of the dislocations around carbide precipitates raises the geometrically necessary dislocation density, resulting in higher KAM values.

### 3.3. Hardness and Tensile Property with Respect to Post-Heat Treatment

Figure 12 shows the Vickers hardness profile of the LP-DED FSS 430 on MSS 410 base metal with various heat treatments. The red dashed line at 0 mm marks the interface between the deposited FSS 430 and base metal MSS 410. In the as-built condition, hardness on the MSS 410 side sharply increased near the interface, peaking at approximately 480 HV_1.0_, while the MSS 410 side exhibited a lower hardness of around 200 HV_1.0_. This sharp transition indicates the presence of a heat-affected zone (HAZ) on the MSS 410 side, resulting from rapid cooling during the additive manufacturing process. In the SHT-815 sample, the peak hardness near the interface was reduced, resulting in softening due to stress relief and partial tempering. The hardness on the FSS 430 side remained relatively unchanged. For the SHT-980 sample, the highest hardness of 480–500 HV_1.0_ was observed in the MSS 410 region, which was attributed to chromium carbide precipitation followed by phase transformation. However, the FSS 430 still showed a modest hardness variation due to its ferritic nature, which did not harden significantly through heat treatment.

Figure 13 shows the tensile properties at 500 °C for the LP-DED FSS 430 on MSS 410 base metal with various post-heat treatments. Although all samples showed similar tensile behavior, the yield strength, tensile strength, and elongation of the as-built sample at the scan speed of 1100 mm/min were 360 MPa, 545 MPa, and 19%, respectively. These as-built tensile properties indicate slightly higher values compared to the laser-welded stainless steel AISI 430 by Evin et al. [51]. Compared to the as-built sample, it reveals a modest increase in tensile strength with 540 to 560 MPa depending on post-heat treatment temperature. Among the solution heat-treated samples, those heat-treated at 980 °C retained higher tensile strength than those heat-treated 815 °C. This could be attributed to the grain refinement from prior LP-DED microstructure and chromium carbide coarsening at higher heat treatment, which can increase dislocation density and yield harder mechanical behavior, as shown in Figure 10 and Figure 11. In contrast, the SHT-815 may not have been heat-treated high enough to trigger full recrystallization, leaving a less uniform and softer microstructure. It is revealed that the higher tensile strength of the SHT-980 was closely correlated with its higher Vickers hardness and microstructural evolution, including chromium carbide coarsening, which enhances dislocation pinning and suppresses plastic deformation, leading to superior tensile performance.

## 4. Conclusions

FSS 430 deposits on MSS 410 base metal were produced by laser powder-directed energy deposition, and then the samples underwent various post-heat treatments. The following conclusions can be drawn from this study:(a)The three major ferrite peaks with BCC structures (110), (200), and (211) were observed in all of the samples of as-built, SHT-815, and SHT-980.(b)Irregular ferrite and tiny chromium carbides throughout the matrix of the as-built sample were observed. The prior chromium carbide (Cr_23_C_6_) precipitates partially dissolved into the ferrite matrix, and many small carbides remained in the grain and at the grain boundaries in the samples of SHT-815. Meanwhile, the prior chromium carbide precipitates were dissolved completely and precipitated to coarser chromium carbides or Ostwald ripening along the grain boundaries in the samples of SHT-980.(c)The highest misorientation (KAM = 0.37°) with solution heat treatment at 980 °C is mostly attributed to significant chromium carbide coarsening in the region of AM FSS 430 deposits. Most strain intensively occurred in the vicinity of the joint interface between the AM FSS 430 deposit and MSS 410. For the as-built sample, more strain was concentrated beneath the joint interface and in the heat-affected zone. In the SHT-815 interface, most strain was concentrated along the interface. The highest strain was observed at the coarsened chromium carbides above the interface, beneath the joint interface, and in the heat affected zone in the samples of SHT-980.(d)The highest Vickers hardness of 480–500 HV_1.0_ was observed in the MSS 410 region, which was attributed to chromium carbide precipitation followed by phase transformation. However, the FSS 430 still showed a modest hardness variation due to its ferritic nature, which did not harden significantly through heat treatment.(e)Compared to the as-built sample, the results reveal a modest increase in tensile strength with 540 to 560 MPa depending on the post-heat treatment temperature. Among the solution heat-treated samples, those heat-treated at 980 °C retained higher tensile strength than those heat-treated at 815 °C.

## Figures and Tables

**Figure 1 micromachines-16-00494-f001:**
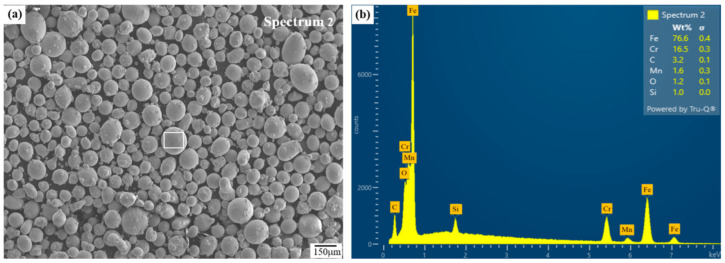
Micrographs of (**a**) feedstock FSS 430 powder and (**b**) EDS analysis.

**Figure 2 micromachines-16-00494-f002:**
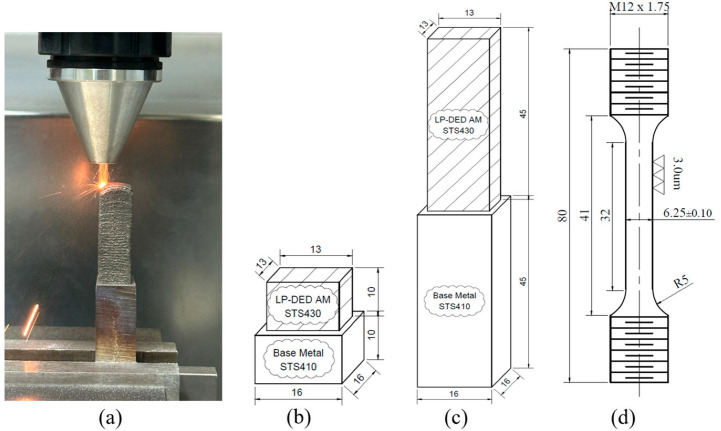
Preparations for (**a**) deposition of tensile specimen, drawings of (**b**) microstructure and (**c**) tensile samples, and (**d**) tensile specimen size.

**Figure 3 micromachines-16-00494-f003:**
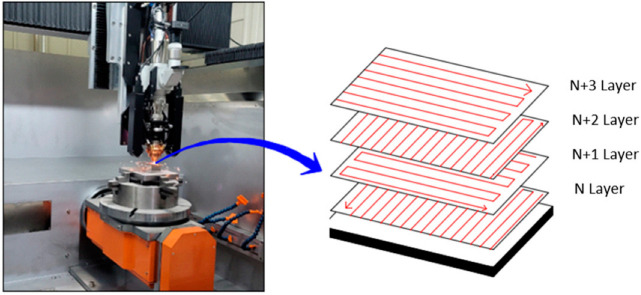
Schematic illustration of directed energy deposition scan strategy.

**Figure 4 micromachines-16-00494-f004:**
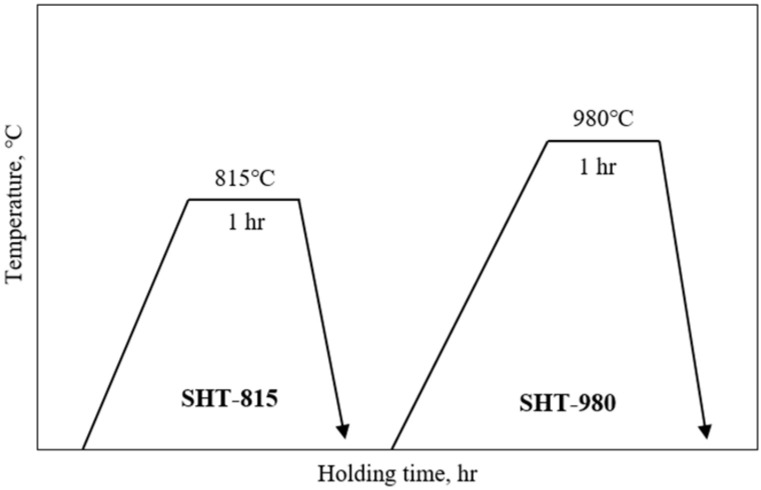
Post-heat treatment for the LP-DED FSS 430 on MSS 410 base metal [28].

**Figure 5 micromachines-16-00494-f005:**
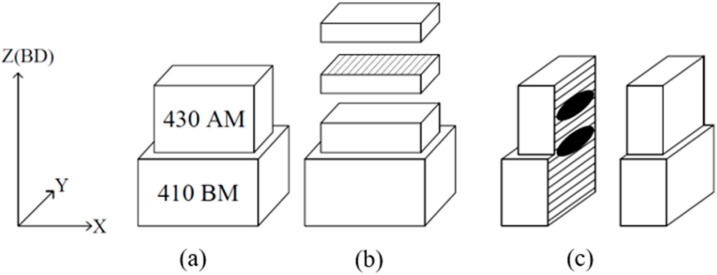
Schematic of (**a**) deposit, (**b**) XRD specimen, and (**c**) metallographic observation.

**Figure 6 micromachines-16-00494-f006:**
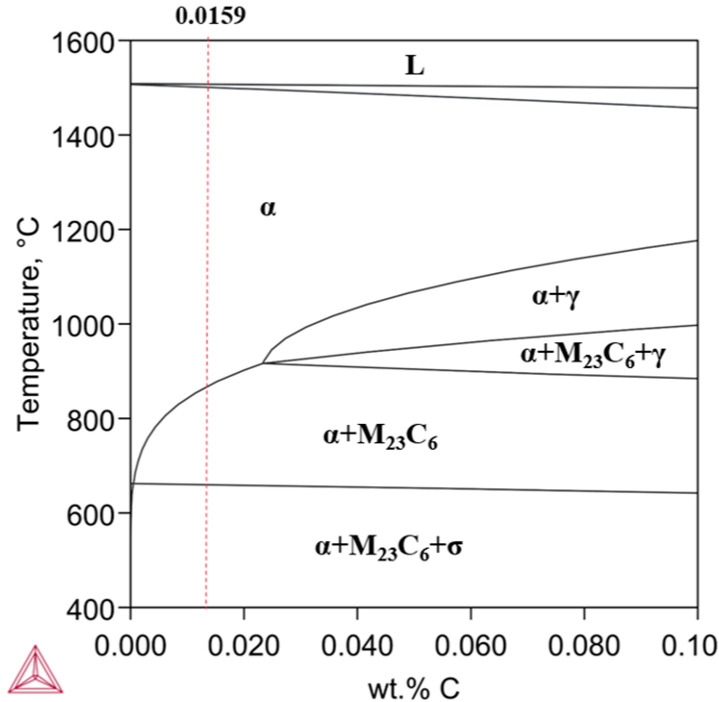
Fe-17.3492Cr-0.1516Ni-0.8271Si-0.9302Mn-xC phase diagram: α-ferrite; γ-austenite; M_23_C_6_-Cr_23_C_6_; σ-sigma.

**Figure 7 micromachines-16-00494-f007:**
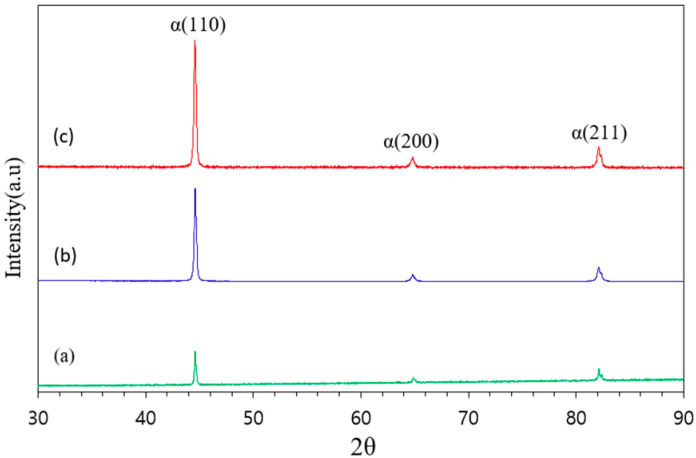
XRD patterns of LP-DED FSS 430 on MSS 410 base metal: (**a**) FSS 430 powder; (**b**) as-built; (**c**) solution treated at 980 °C for 1 h followed by forced fan cooling.

**Figure 8 micromachines-16-00494-f008:**
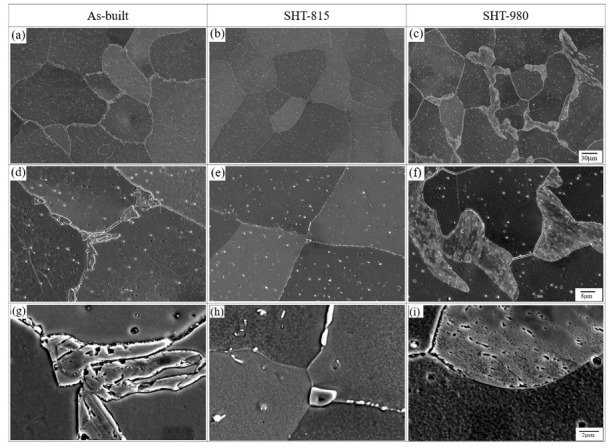
SEM images of LP-DED AM FSS 430 under various post-heat treatments: SHT-815 solution heat-treated at 815 °C for 1 h, followed by forced fan cooling; SHT-980 solution heat-treated at 980 °C for 1 h, followed by forced fan cooling. (**a**,**d**,**g**) As-built; (**b**,**e**,**h**) SHT-815; (**c**,**f**,**i**) SHT-980.

**Figure 9 micromachines-16-00494-f009:**
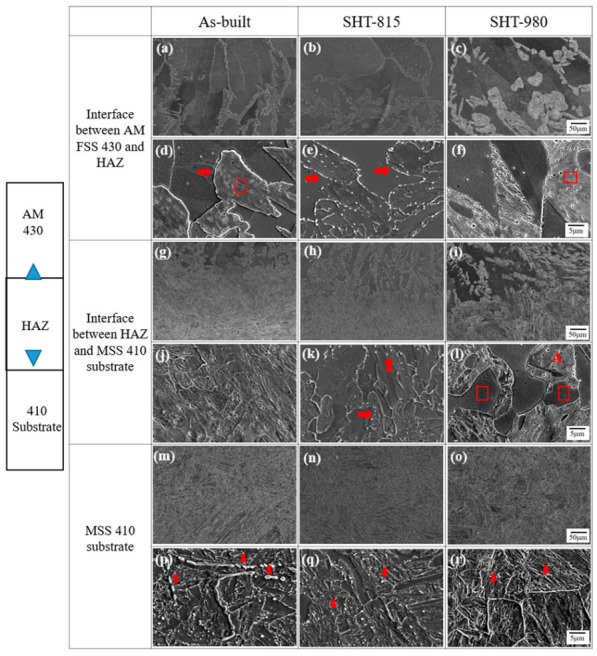
Interfaces between LP-DED FSS 430 and MSS 410 base metal with various post-heat treatments: (**a**,**d**,**g**,**j**,**m**,**p**) As-built; (**b**,**e**,**h**,**k**,**n**,**q**) SHT-815T; (**c**,**f**,**i**,**l**,**o**,**r**) SHT-980.

**Figure 10 micromachines-16-00494-f010:**
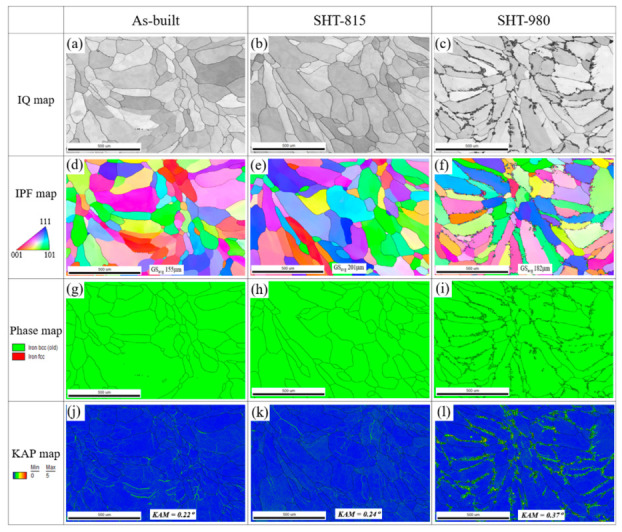
EBSD images of perpendicular (YZ) planes of LP-DED FSS 430 on MSS 410 base metal with various heat treatments: (**a**,**d**,**g**,**j**) As-built; (**b**,**e**,**h**,**k**) SHT-815; (**c**,**f**,**i**,**l**) SHT-980.

**Figure 11 micromachines-16-00494-f011:**
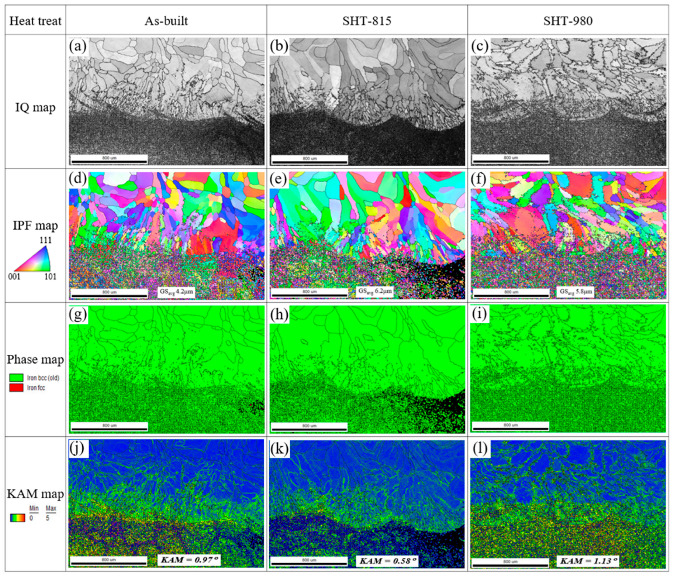
EBSD images of interfaces of LP-DED FSS 430 on MSS 410 base metal with various post-heat treatments: (**a**,**d**,**g**,**j**) As-built; (**b**,**e**,**h**,**k**) SHT-815; (**c**,**f**,**i**,**l**) SHT-980.

**Figure 12 micromachines-16-00494-f012:**
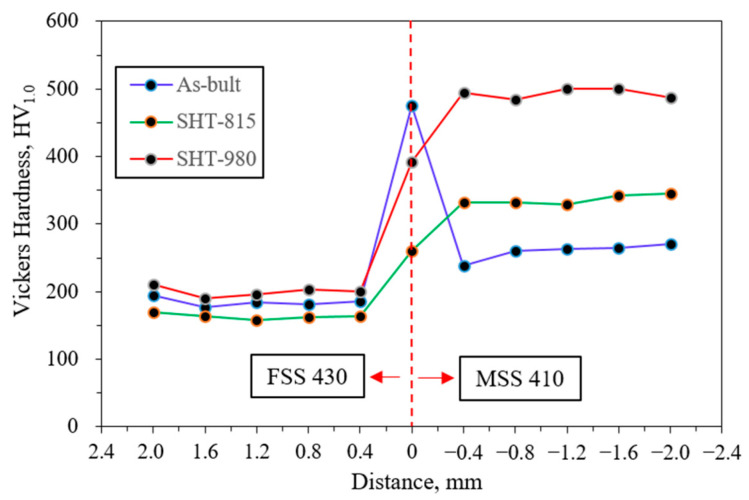
Vickers hardness of LP-DED FSS 430 on MSS 410 base metal with various heat treatments.

**Figure 13 micromachines-16-00494-f013:**
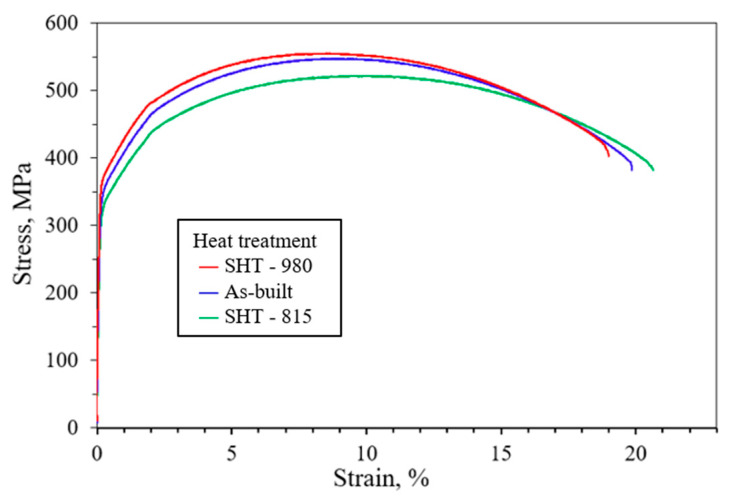
Tensile properties at 500 °C for LP-DED FSS 430 on MSS 410 with various post-heat treatments.

**Table 1 micromachines-16-00494-t001:** Composition of powder, deposit, and base metal (wt.%).

	C	Si	Mn	P	S	Ni	Cr	Fe
AISI 430	<0.12	max. 1.0	max. 1.0	max. 0.04	max. 0.03	0–0.75	16–18	Bal.
Powder	max. 0.08	max. 1.0	max. 1.0	max. 0.04	max. 0.02	-	16–18	Bal.
Deposit	0.0159	0.827	0.930	0.01	0.007	0.152	17.3	Bal.
MSS 410	0.1463	0.3825	0.605	0.02	0.003	0.018	12.2	Bal.

**Table 2 micromachines-16-00494-t002:** Deposition parameters of the LP-DED MSS 430.

Process	Parameter	Value
	Laser power (W)	600
	Scanning speed (mm/min)	1100
LP-DED	Powder feed rate (g/min)	7
	Hatch distance (mm)	0.5
	Layer thickness (mm)	0.45

**Table 3 micromachines-16-00494-t003:** Peak data and lattice parameters.

	Miller Indices	2θ (°)	θ (°)	d-Spacing (Å)	a (Å) = d·√(h^2^ + k^2^ + l^2^)
Before Heat Treatment	(110)	44.6162	22.3081	2.03099	2.03099 × √2 = 2.872 Å
	(200)	64.8096	32.4048	1.43859	1.43859 × √4 = 2.877 Å
	(211)	82.1571	41.0786	1.17325	1.17325 × √6 = 2.872 Å
	Average Lattice Parameter (As-Built FSS 430)				≈2.874 Å
After Heat Treatment	(110)	44.5913	22.2957	2.03038	2.03038 × √2 = 2.871 Å
	(200)	64.7939	32.3970	1.43771	1.43771 × √4 = 2.877 Å
	(211)	82.0942	41.0471	1.17302	1.17302 × √6 = 2.872 Å
	Average Lattice Parameter (Heat-Treated at 980 °C FSS 430)				≈2.873 Å

## Data Availability

The data that support the findings of this study are available from the corresponding author upon reasonable request.

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
