# Peer review of "High-Temperature Properties of LP-DED Additive Manufactured Ferritic STS 430 Deposits on Martensitic STS 410 Base Metal"

_micromachines, 2025, doi:10.3390/mi16050494_

Round 1
Reviewer 1 Report
Comments and Suggestions for Authors
This study investigates the microstructural evolution, grain growth, carbide precipitation near the interface, and high-temperature tensile properties of LP-DED additive-manufactured ferritic stainless steel (FSS) 430 deposits on martensitic stainless steel (MSS) 410 base metal under various post-heat treatments. The research provides valuable insights into the behavior of these materials under high-temperature conditions, which is crucial for industrial applications.
However, some aspects of the study could be improved or expanded upon.
Lines 184-196 – the XRD analysis does not provide the lattice parameters of the detected phases and their changes.
For full certification of the material, there is not enough data on hardness and density/porosity.
Why have no direct studies of corrosion resistance been conducted?
Line 376 - Check the necessity of the phrase at the beginning of the Funding section
Author Response
Comments and Suggestions for Authors
This study investigates the microstructural evolution, grain growth, carbide precipitation near the interface, and high-temperature tensile properties of LP-DED additive-manufactured ferritic stainless steel (FSS) 430 deposits on martensitic stainless steel (MSS) 410 base metal under various post-heat treatments. The research provides valuable insights into the behavior of these materials under high-temperature conditions, which is crucial for industrial applications.
However, some aspects of the study could be improved or expanded upon.
Q1) Lines 184-196 – the XRD analysis does not provide the lattice parameters of the detected phases and their changes.
A1) Revised and please refer to new Table 3 and revised text Lines 193-203.
Q2) For full certification of the material, there is not enough data on hardness and density/porosity.
A2) The chemical composition of MSS 410 has been corrected and highlighted in Table 1.
Hardness measurements are presented in Figure 12 and discussed in Lines 335-347, 374-379, 405-408.
Due to time constraints, density and porosity were not assessed in this study. We appreciate your constructive
recommendation and will consider these evaluations in future works.
Q3) Why have no direct studies of corrosion resistance been conducted?
A3) This study specifically focused on the high-temperature tensile properties of LP-DED additive-manufactured
FSS 430 deposits on MSS 410.
We acknowledge the importance of corrosion resistance and plan to address this in a future investigation. Thank you for your insightful suggestion, which will guide our subsequent experimental work
Q4) Line 376 - Check the necessity of the phrase at the beginning of the Funding section
A4) The phrase 'Please add:' at the beginning of the Funding section (Line 421) has been removed as advised

Reviewer 2 Report
Comments and Suggestions for Authors
The work presents an experimental investigation on LP-DED additively manufactured ferritic STS 430 deposits on martensitic STS 410 substrate. The authors have undergone a metallurgical evaluation on the properties of the deposit. The significant content of the manuscript is sound and convincing. There is a flow in writing in the paper and the results are well-presented. There are some minor comments:
- The authors are advised to have a spell-check before the submission.
- Figure 1 may contain the elemental composition as well.
- Figure 4 needs citation or the scale is missing on the axes.
- I could see obsolete citations. I would strongly suggest to add papers from last five years to maintain the quality of the journal.
Author Response
Comments and Suggestions for Authors
The work presents an experimental investigation on LP-DED additively manufactured ferritic STS 430 deposits on martensitic STS 410 substrate. The authors have undergone a metallurgical evaluation on the properties of the deposit. The significant content of the manuscript is sound and convincing. There is a flow in writing in the paper and the results are well-presented. There are some minor comments:
Q1) The authors are advised to have a spell-check before the submission.
A1) Done.
Q2) Figure 1 may contain the elemental composition as well.
A2) Figure 1(b) has been revised to include the chemical composition in the upper right box.
Line 89 has also been edited accordingly.
Q3) Figure 4 needs citation or the scale is missing on the axes.
A3) A citation has been added as reference [28]. The scale is not missing, as it is not required for this figure.
Q4) I could see obsolete citations. I would strongly suggest to add papers from last five years to maintain the quality
of the journal.
A4) The references have been updated, and recent publications have been highlighted for your review.
